# Long-Term Exposure of Cultured Astrocytes to High Glucose Impact on Their LPS-Induced Activation

**DOI:** 10.3390/ijms25021122

**Published:** 2024-01-17

**Authors:** Ayna Abdyeva, Ekaterina Kurtova, Irina Savinkova, Maksim Galkov, Liubov Gorbacheva

**Affiliations:** 1Faculty of Medical Biology, Pirogov Russian National Research Medical University of the Ministry of Health of the Russian Federation, 117997 Moscow, Russia; nynaynab@gmail.com (A.A.); eekurtova@gmail.com (E.K.); irenesavik@mail.ru (I.S.); galkovm@yandex.ru (M.G.); 2Faculty of Biology, Lomonosov Moscow State University, 119991 Moscow, Russia

**Keywords:** diabetes mellitus, β-hexosaminidase, hyperglycemia, lipopolysaccharide (LPS), astrocyte, inflammation, Interleukin 6 (IL-6), nitric oxide (NO)

## Abstract

Diabetes mellitus is associated with various complications, mainly caused by the chronic exposure of the cells to high glucose (HG) concentrations. The effects of long-term HG exposure in vitro accompanied by lipopolysaccharide (LPS) application on astrocytes are relatively unknown. We used cell medium with normal (NG, 5.5 mM) or high glucose (HG, 25 mM) for rat astrocyte cultures and measured the release of NO, IL-6, β-hexosaminidase and cell survival in response to LPS. We first demonstrated that HG long-term incubation of astrocytes increased the release of β-hexosaminidase without decreasing MTT-detected cell survival, suggesting that there is no cell membrane damage or astrocyte death but could be lysosome exocytosis. Different from what was observed for NG, all LPS concentrations tested at HG resulted in an increase in IL-6, and this was detected for both 6 h and 48 h treatments. Interestingly, β-hexosaminidase level increased after 48 h of LPS and only at HG. The NO release from astrocytes also increased with LPS application at HG but was less significant. These data endorsed the original hypothesis that long-term hyperglycemia increases proinflammatory activation of astrocytes, and β-hexosaminidase could be a specific marker of excessive activation of astrocytes associated with exocytosis.

## 1. Introduction

Diabetes is a group of diseases associated with metabolic disorders due to a chronic increase in blood glucose levels. A persistent increase in blood glucose level, or hyperglycemia, contributes to the development of pathological changes in the body. The most dangerous consequences of the development of diabetes are systemic vascular complications, including nephropathy, retinopathy, and damage to the large vessels of the heart, brain, and arteries of the lower extremities [1]. These complications are the main causes of disability and mortality in patients with diabetes. The progression of pathological changes in the body during the development of diabetes is associated with both the direct toxic effect of chronic hyperglycemia and the activation of the immune system [2,3]. During hyperglycemia, damage to target organs occurs as a result of the activation of multiple metabolic pathways that provide excessive glucose shunting [4]. Under conditions of hyperglycemia, the polyol pathway of glucose oxidation is activated, in which glucose is converted into osmotically active sorbitol and fructose, while NADPH, which is of great importance in the functioning of NO-synthase and the antioxidant systems of the cell, is consumed [5].

In recent years, it has become obvious that hyperglycemia accompanying diabetes leads to neuroinflammatory processes, neurodegeneration and general dysfunction of the central nervous system. Diabetic encephalopathy is characterized by cognitive and motor disorders associated with changes in metabolic processes in the central nervous system, increasing vascular dysfunction and the development of oxidative stress [6]. Cognitive changes observed in patients with diabetes are associated with changes in the ratio of the volumes of white and grey matter. Long-term hyperglycemia leads to a decrease in the volume of grey matter in the thalamus, temporal lobes, and parahippocampal gyrus, i.e., the brain areas responsible for the implementation of higher brain functions, including memory and attention [7]. In experimental animal models of diabetes, tests to determine the effectiveness of associative learning showed that hyperglycemia led to long-term negative changes in synaptic plasticity in the hippocampus. Researchers explained the observed deterioration in declarative memory by the toxic effect of hyperglycemia on hippocampal neurons [7]. It is assumed that the observed impairment of synaptic plasticity under DM modeling conditions is associated with an impairment in the efficiency of neurotransmitter release into the synaptic cleft [8,9]. Astrocytes serve as a source of energy substrates and ATP, including the area of synaptic contacts. 

Astrocytes play a crucial role in the development of CNS insults through a process known as reactive astrogliosis [10]. This response involves changes in astrocyte morphology and proliferation. Morphological changes are accompanied by increased expression of the genes and proteins, which differs from the ones expressed in a resting state. In vitro experiments on cell cultures grown in a high-glucose medium and on brain slices from rats with diabetes showed reduced interaction between astrocytes through the gap junctions [11]. In vitro, high glucose levels (30 mM/L) resulted in changes in the expression of cytoskeleton proteins such as GFAP and vimentin [12]. Additionally, in astrocytes, high glucose levels can also induce the expression of inflammatory cytokines such as tumor necrosis factor-α (TNF-α), interleukin-6 (IL-6), interleukin-1 (IL-1) and interleukin-4 (IL-4) [13,14]. These cytokines are signaling molecules that can promote inflammation and contribute to the progression of diabetic cerebral neuropathy. Chronic hyperglycemia evokes changes in the vascular endothelial growth factor (VEGF) secretion, affecting the endothelial cells’ migration, proliferation and angiogenesis [15]. The long-term cultivation of cells in a high-glucose medium leads to the activation of the AMP-activated protein kinase (AMPK) signaling pathway, decreased metabolic efficiency and capacity, and increased glucose uptake and glycogen storage [16,17]. Moreover, it was shown that a drug for the treatment of diabetes, metformin, exhibits anti-inflammatory effects on LPS-activated primary rat astrocytes cultured in a medium with a high glucose concentration [18].

One of the effects of high glucose on astrocytes is the increased production of reactive oxygen species (ROS) and advanced glycation end-products (AGE). ROS are highly reactive molecules that can cause damage to cellular components, leading to oxidative stress and inflammation. This oxidative stress can further worsen the damage caused by diabetes in the CNS. AGE may exert proinflammatory effects through activating receptors for advanced glycation end-products (RAGE) and some pattern-recognizing receptors (PRR), thus promoting neuroinflammation [9].

The high glycolytic capacity of brain tissue is presumably due to the activity of astrocytes. They process glucose through aerobic glycolysis, as indicated by a specific set of active genes, enzyme proteins and a small number of mitochondria in astrocytes [19]. Chronic hyperglycemia in diabetes mellitus leads to increased permeability and even dysfunction of the blood–brain barrier (BBB). Hyperglycemia-dependent oxidative stress can activate an innate immune hyperglycemia-dependent inflammatory response within the BBB, which can disrupt its integrity and increase permeability [20]. The state of astrocytes determines the integrity and functional state of the BBB. Changes in the reactivity of astrocytes influence the permeability of the BBB and trigger the transformation of acute brain conditions into chronic ones [21]. Astrocytes modulate diabetic BBB integrity via glial-derived neurotrophic factor, nitric oxide, tumor necrosis factor α, reactive oxygen-nitrogen, advanced glycation end-products and some other biologically active substances [12].

Thus, the functional activity of astrocytes is disrupted during the development of hyperglycemia. Therefore, the understanding of the mechanisms of astrocyte influences on other brain cells in hyperglycemia and the search for key molecular markers of pathological cell states are required for the pharmacological prevention of neuroinflammation and cell injury.

To our knowledge, there are no data on the long-term influence of high glucose in primary cultured astrocytes exposed to lipopolysaccharide (LPS). Therefore, the aim of the study was to assess the effect of normal/high glucose on the proinflammatory activity of primary astrocytes induced by treatment with different concentrations of LPS.

## 2. Results

### 2.1. The Morphology of Astrocytes Cultured in Normal- or High-Glucose Medium

Primary rat astrocytes were cultured in DMEM containing a normal (5.5 mM, NG) or high (25 mM, HG) concentration of glucose according to the experimental protocol (Figure 1). Astrocytes showed changing morphology after 14 DIV cultivation in HG medium (Figure 2). The immunocytochemistry for GFAP demonstrated that astrocytes cultured in NG medium exhibited a flat polygonal shape, whereas astrocytes cultured in HG medium showed an atypical appearance, elongated shape with long branches and with low number cell–cell contacts (Figure 2).

### 2.2. Cell Survival and NO Production of Rat Astrocytes Cultured in Normal- or High-Glucose Medium

The survival of astrocytes was measured after the 48 h cultivation of astrocytes in the serum-free medium containing NG or HG using the MTT test (Figure 1). Cell survival was 8.5% higher in the HG medium as compared to astrocytes cultured in the NG medium (Figure 3A). 

The iNOS expression in glial cells and macrophages results in high levels of NO and peroxynitrite, which ultimately cause damage to the CNS [22]. Therefore, the level of nitrites in the culture medium can be considered a result of astrocyte activation.

In this study, we measured the level of nitrite by the Greiss reaction as an indicator of NO production. The NOx level was measured after the 48 h incubation of cells in the HG serum-free (Figure 1). Data presented in Figure 3C show that the nitrite production did not change under the HG conditions in comparison to the NG conditions.

### 2.3. Effect of LPS on NO Production by Astrocytes Exposed to Normal or High Glucose

The main symptom of diabetes mellitus is hyperglycemia, which refers to elevated levels of glucose in the blood plasma, which correlates with an increased incidence of innate immunity-dependent low-grade systemic inflammation [20,23,24,25]. The proinflammatory agent LPS is an agonist of the toll-like receptor-4 (TLR4). TLR4 is a member of the family of TLR receptors, which are the important targets of the glial innate immune system. The receptor activation and the respective signaling pathways in astrocytes lead to the production of pro- and anti-inflammatory cytokines and polyunsaturated fatty acid derivatives such as prostaglandins [26,27].

In the present study, we estimated the influence of glucose concentrations on the viability and NO production of primary astrocytes during the application of LPS (Figure 3B,D).

Cell culturing of the rat brain astrocytes for 48 h with LPS (0.1, 1 µg/mL) did not alter cell survival in either the NG or HG mediums (Figure 3B).

Then, we measured the levels of nitrites in the culture medium, as the marker of NO production in LPS-treated glial cultures (Figure 1). LPS dose-dependently stimulated NO accumulation in astrocyte cultures (Figure 3D). Furthermore, 48 h incubation of astrocytes with 0.1 and 1 µg/mL LPS showed that the nitrite accumulation was observed only in the presence of 1 µg/mL LPS. Moreover, this increase was significantly lower in the HG conditions as compared to the NG conditions (Figure 3D). It is important to note that LPS at a concentration of 1 mg/mL significantly increased nitrite production in astrocytes after the 48 h incubation in both the NG and HG conditions (Figure 3D).

### 2.4. Effect of LPS on IL-6 Release in Astrocytes Exposed to Normal or High Glucose

The neuroinflammation response is a multistep process characterized in particular by the release of signaling molecules. In this regard, in our experiments, we examined the IL-6 release by astrocytes.

We exposed astrocyte cultures to 0.1 or 1 µg/mL LPS for 6 or 48 h in NG or HG conditions. IL-6 was practically undetectable in the culture medium under the control conditions when the astrocytes were cultivated in the NG medium at both time points (6 h and 48 h) (Figure 4). LPS application to astrocytes in the NG medium did not result in IL-6 accumulation either. However, we were able to measure IL-6 when the astrocytes were exposed to LPS in the HG medium for both 6 h and 48 h (Figure 4).

Figure 4 shows the LPS concentration-dependent IL-6 increase in the medium from astrocytes cultivated under the HG conditions.

### 2.5. Effect of LPS on the β-Hexosaminidase Release by Astrocytes Exposed to Normal or High Glucose

Previously, the release of β-hexosaminidase was known as an index of mast cell degranulation. Some authors have analyzed the proinflammatory effect of substances based on their possibility to release not only IL-1 but also β-hexosaminidase [28]. Here, we used this test as a marker of astrocyte activation.

The cells were stimulated with LPS (0.1 and 1 µg/mL) for 6 and 48 h. We examined β-hexosaminidase levels in the extracellular space to estimate lysosome exocytosis. Our experiments showed that the incubation of astrocytes with 0.1 or 1 µg/mL LPS for 6 h did not affect the level of extracellular β-hexosaminidase independently of the HG or NG conditions (Figure 5). We found increased β-hexosaminidase activity after the 48 h exposure of astrocytes to LPS in the HG medium (Figure 5). We have to note that HG affected the β-hexosaminidase level in the medium (Figure 5) because its activity was more than 2-fold higher as compared to the NG when it was studied at 48 h (Figure 5). 

Earlier, the possibility of astrocytes boosting lysosome exocytosis without damaging cell membranes was shown [29]. Here, we demonstrate this effect in cultured astrocytes after LPS- and HG-induced activation. Taken together, these results and MTT test data indicate that high glucose alone or with LPS treatment induced lysosome exocytosis, as detected by the release of β-hexosaminidase in astrocyte primary culture.

## 3. Discussion

In the present study, we demonstrated that HG long-term incubation (14–16 days) of primary astrocytes increased the release of the proinflammatory cytokine IL-6 and the lysosomal enzyme β-hexosaminidase and changed cells’ morphology.

Hyperglycemia is one of the major symptoms seen in patients with diabetes. DM is one of the most prevalent chronic diseases, and it is an important public health problem. The International Diabetes Federation (IDF) informs that 537 million adults are now living with diabetes worldwide, and there is a rise of 16% (74 million) since the previous IDF estimates in 2019 [https://idf.org/news/diabetes-now-affects-one-in-10-adults-worldwide/, accessed on 15 December 2023]. When the body cannot produce or use insulin effectively, that leads to high blood glucose levels (hyperglycemia). Long-term high glucose levels are associated with damage to the body and the failure of various organs and tissues. Criteria for the diagnosis of diabetes include fasting plasma glucose levels ≥ 7.0 mM, plasma glucose levels in 2 h after receiving a 75 g glucose load ≥ 11.1 mM, and/or casual (or random) plasma glucose levels ≥ 11.1 mM [30]. Normal type is defined as fasting plasma glucose levels of 5.6 mM and plasma glucose levels in 2 h after receiving a 75 g glucose load of 7.8 mM [30]. Taking this into account, we suppose that the glucose concentration in NG (5.5 mM) DMEM is similar to the normal human plasma glucose level, whereas the HG (25 mM) glucose level is 2-fold higher than the diagnostic values of human plasma glucose at diabetes.

The protocol of astrocyte cultivation in 5.5 or 25 mM glucose-containing DMEM (14–16 days), which was applied in the present study, is commonly used as a model for diabetes and for the testing of drugs to reverse or minimize hyperglycemic damage [11].

Our data point to the changes in cellular morphology during the 14-day cultivation of astrocytes in HG medium in contrast to NG. These effects of HG are consistent with data from other researchers indicating a change in the expression of cytoskeleton proteins such as GFAP and vimentin at high glucose levels (30 mM/L) [12]. Moreover, in vitro experiments on cell cultures grown in a high-glucose medium and on brain slices from rats with diabetes showed reduced interaction between astrocytes through the gap junctions [11]. In our experiments, we have shown that the flat polygonal shape of astrocytes cultured in NG medium that form a continuous cell monolayer changes by cell culturing in HG medium. Astrocytes have an elongated shape with long branches and a low number of cell-cell contacts at HG (Figure 2).

Our results of the MTT test indicated that the long-term (16 days) cultivation of primary astrocytes at high glucose leads to an 8.5% increase in cell survival after 48 h of incubation in the serum-free medium compared to astrocytes cultured in the medium with normal glucose. Under similar conditions but during a shorter 24 h exposure and with serum in the medium, primary astrocyte proliferation was significantly lower (about 25%) in a high-glucose medium [31].

The specificity of the MTT test as a method for determining the level of dehydrogenases in cells suggests that an increase in the level of tetrazolium salts can be considered not only as an increase in cell proliferation/survival but also as cellular hyperplasia. Therefore, it is difficult to compare data from different studies on the cell cultures, which are characterized by various conditions such as duration of cultivation, presence of serum in the medium, etc. Our data indicate a slight increase in survival/hyperplasia of rat astrocytes that were cultured for 14 days in a high glucose medium, with the following 48 h in a serum-free medium (Figure 3A). The ambiguity of the effects of high glucose on cell survival is indicated by numerous studies obtained on different types of cells and under different experimental conditions.

The high-glucose conditions affect cell proliferation, morphology, apoptosis and protein expression in nonneural cells, such as endothelial, mast, macrophage and smooth muscle cells [32,33,34,35,36]. Along with the toxic effect of high glucose on different types of cells (endotheliocytes, etc.), there are some data on the HG protective effect of HG. Thus, the medium with HG (15–60 mM/L glucose for 24–30 h) reduced the hypoxic injury of primary astrocytes [37]. The most interesting data are those about how glucose affects the activation of astrocytes, which is seen during brain damage and is thought to be part of the inflammatory response. A number of previous studies have reported the influence of high glucose on glial morphology and reactivity, both in vitro and in vivo [11,14,31,38]. In experiments in vivo, streptozotocin-induced diabetes in mice led to significantly increased activation of microglia and astroglia, the loss of neurons, and cognitive dysfunctions [38]. At the same time, a moderate increase in glucose, on the other hand, decreased the level of markers of astrocyte activation. Astrocytes had a reduced content of S100B and glial fibrillary acidic protein when cultured in a medium with 12 mM glucose, as well as a reduced content of glutathione and cell proliferation rate [31]. Recent data indicate the glucose-induced proinflammatory states in astrocytes cultivated in a medium with high glucose, which is confirmed an increase in the amounts of free polyunsaturated fatty acids, such as arachidonic, docosahexaenoic and eicosapentenoic acids, and cyclooxygenase-mediated metabolites [14].

The activation of astrocytes and triggering of astrogliosis is a step of the neuroinflammation response, which is characterized by the sequential activation of signaling molecules and transcription factors. Astrocyte production of tumor necrosis factor (TNF), interleukin 6 (IL-6) and nitric oxide (NO) has been implicated in increased inflammatory status and secondary damage [39,40,41], which is partly mediated by TLR4. 

It is known that TLR4 mediates the neuroinflammatory effects of LPS in cultured astrocytes, and its expression remarkably increases in diseased CNS and in reactive astroglia [42,43]. LPS interaction with TLR4 activates downstream mitogen-activated protein kinase (MAPK) and NF-B signaling pathways and subsequently causes inflammatory mediator production [41,42,43,44]. Here, we studied some LPS-activated pathways in a cellular model of diabetes.

The LPS-induced expression of IL-6 is mediated by NF-κB, a potent immediate early transcriptional regulator of numerous proinflammatory genes [44]. Interestingly, a higher proportion of astrocytes is involved in IL-6 production in vitro in comparison to microglia [41].

Thus, IL-6 plays a key role in proinflammatory immune cell activation, including LPS-dependent activation of astrocytes. Based on this special status of IL-6, we assessed the effect of HG on LPS-induced expression of IL-6 in primary astrocyte cultures. In the experiments performed in cocultures of astrocytes and microglia and in isolated cells, astrocytes were more involved in the production of IL-6 in response to the LPS treatment than in the production of NO and TNF [41]. Our result suggests that long-term hyperglycemia for 14–16 days promotes the activation of rat astrocytes and the release of IL-6 after LPS stimulation at both 6 h and 48 h exposure to endotoxin (Figure 4). We have to note that in our experiments, long-term cultivation of astrocytes in the HG medium simulated diabetes mellitus, a disease with stable hyperglycemia as a main symptom and key damage factor.

In the present study, we have investigated NO production by astrocytes during the application of LPS and at HG in a medium. Early on, it was shown that the highly reactive free radical gas NO had been implicated in the pathogenesis of several diseases, including diabetes [45,46,47]. It has been demonstrated that in glial cells, NO production is mediated by activation of the MAPKs and NF-kappaB pathways, where the ERK and p38 MAP kinase cascades play key roles in the regulation of iNOS gene expression in endotoxin-activated cells [48,49,50,51]. Here, we have shown that 48 h incubation of astrocytes with LPS led to nitrite accumulation in the presence of 1 µg/mL LPS and did not depend on glucose levels in a medium. Taken together, these data and the results of the IL-6 study suggest that IL-6 is a more specific marker of astrocyte activation by LPS and high glucose than NO.

In this study, we have tested a relatively new marker of astrocyte activation: the release of β-hexosaminidase. β-hexosaminidase is a lysosomal hydrolase that removes terminal N-acetylgalactosamine from the GM2 ganglioside. 

Lysosomal enzymes, secreted following rupture of the lysosomal membrane, can be very harmful to their environment and result in the pathological destruction of cellular structures. On the other hand, the secretion and release of lysosomal components from cells may serve as markers of cell activation and damage [52]. For example, a high plasma level of hexosaminidase A is a marker of enterocyte necrosis [53]. Lysosomal enzymes appeared in the infarct zone due to cell destruction and could participate in delayed cell death during brain ischemia [52]. Thus, β-hexosaminidase is a kind of lysosomal enzyme that would leak under pathological conditions during lysosome exocytosis and demonstrates the terminal stage of cell activation [29,54]. In accordance with this, some authors have analyzed the proinflammatory effect of substances based on their possibility to release not only IL-1 but also β-hexosaminidase [28].

The possibility of using hexosaminidase as a marker of astrocyte activation during hyperglycemia is also indicated by the fact that exposure of the LAD2 human mast cell line to high glucose-induced intracellular and extracellular β-hexosaminidase activities [33].

Moreover, mast cell degranulation, as assessed by β-hexosaminidase release, needs protein kinase C (PKC) activation [55], which was detected in neurons of rats with diabetes mellitus [56]. It is, hence, plausible that the β-hexosaminidase can be a marker of diabetes-induced activation of astrocytes.

Our experiments showed that the incubation of astrocytes with 0.1 or 1 µg/mL LPS for 48 h increased β-hexosaminidase activity in contrast to 6 h exposure (Figure 5). At the same time, β-hexosaminidase activity was more than 2-fold higher in the HG medium as compared to the NG medium (Figure 5). These data point to a rise in β-hexosaminidase activity due to both 48 h exposures of 1 µg/mL LPS and HG. The MTT-test indicated that the increase in extracellular β-hexosaminidase activity was not due to cell necrosis and cell membrane disruption but to the alteration of cell activities, such as lysosome exocytosis.

Increasing evidence shows that lysosomes of astrocytes are involved in ATP release extracellularly in response to different simulations [54,57,58], including trauma, ischemia and other damage to the brain. Moreover, lysosomes have different modes of exocytosis that contribute to intercellular signaling under different pathological conditions [54]. It has recently been shown that H_2_O_2_ can induce lysosome release from astrocytes, which is not associated with damage to cell membranes, and β-hexosaminidase levels were detected in the extracellular space, supporting direct lysosome exocytosis [29]. 

To our knowledge, there are no data about the effect of HG concentrations on β-hexosaminidase release in astrocytes exposed to LPS. Here, we show for the first time that exposure to LPS in the high-glucose conditions stimulated the β-hexosaminidase release in 48 h but not in 6 h after the treatment. Thus, the long-term exposure of astrocytes to high glucose significantly changes LPS-evoked IL-6 and β-hexosaminidase release. These substances trigger the formation of different inflammatory and damage processes and reflect different aspects of astrocyte activation. 

IL-6 may reflect early astrocyte activation in contrast to β-hexosaminidase, which, probably, may be the marker of delayed astrocyte activation associated with exocytosis, leading to a dramatic increase in extracellular ATP. These data endorsed the original hypothesis that long-term hyperglycemia increases proinflammatory activation of astrocytes and that β-hexosaminidase could be a specific marker of excessive activation of astrocytes associated with exocytosis. Thus, our results indicated that LPS-induced proinflammatory activation of astrocytes in vitro rises after long-term (16 days) high glucose exposure, and β-hexosaminidase could be an additional marker of astrocyte activation and a potential target for protecting the brain from injury.

## 4. Materials and Methods

### 4.1. Reagents

Hanks’ balanced salt solution (HBSS) Ca2+ and Mg2+ free ((Gibco, Thermo Fisher Scientific, 168 Third Avenue, Waltham, MA, USA), HBSS with Ca2+ and Mg2+ ((Gibco, Thermo Fisher Scientific, 168 Third Avenue, Waltham, MA, USA), papain (Sigma-Aldrich, St. Louis, MO, USA), DNase (Sigma-Aldrich, St. Louis, MO, USA), DMEM Low glucose (1 g/L) (PanEco, Moscow, Russia), DMEM High glucose (4 g/L) (PanEco, Moscow, Russia), antibiotic-antimycotic ((Gibco, Thermo Fisher Scientific, 168 Third Avenue, Waltham, MA, USA), L-glutamine (Sigma-Aldrich, St. Louis, MO, USA), fetal bovine serum (Invitrogen, Thermo Fisher Scientific, 168 Third Avenue, Waltham, MA, USA), trypsin (Gibco, Thermo Fisher Scientific, 168 Third Avenue, Waltham, MA, USA), 3-(4,5-dimethylthiazol-2-yl)-2,5-diphenyltetrazolium bromide (MTT) (Sigma-Aldrich, St. Louis, MO, USA), Lipopolysaccharide (LPS) (Sigma-Aldrich, St. Louis, MO, USA), Griess reagent (Sigma-Aldrich, St. Louis, MO, USA), RIPA lysis buffer (Sigma-Aldrich, St. Louis, MO, USA), Complete Protease Inhibitor Cocktail (Roche Diagnostics, Mannheim, Germany), Quantikine Rat Il-6, R&D Systems, USA, 4-nitrophenyl-N-acetyl-β-D-glucosaminide (Sigma-Aldrich, St. Louis, MO, USA), Bradford reagent (Sigma-Aldrich, St. Louis, MO, USA).

### 4.2. Primary Astrocyte Cell Culture

All procedures involving animals were performed in accordance with the ethical principles and regulatory documents recommended by the European Convention on the Protection of Vertebrate Animals used for experiments (Guide for the Animals and Eighth Edition. 2010), as well as in accordance with the “Good Laboratory Rules practice”, approved by order of the Ministry of Health of the Russian Federation No. 199n of 4 January 2016. 

Primary astrocyte culture was obtained from the cortex of newborn Wistar rats of both sexes, as described earlier, with some modifications [59,60]. In brief, the animals were anesthetized, decapitated, brains were removed and separated from the meninges. The extracted hippocampi and cortex tissues were washed in a Ca2+ and Mg2+ free Hanks solution (HBSS, (Gibco, Thermo Fisher Scientific, 168 Third Avenue, Waltham, MA, USA), chopped, and placed in a papain solution (papain 500 µg/mL, DNase 10 µg/mL) for 10 min at 37 °C. In order to wash the astrocytes from papain more carefully, Hanks solution (without Ca2+ and Mg2+, DNase 10 µg/mL) was removed several times and added again. Then, the cells were suspended and dispensed through a cell nylon strainer (mesh size 100μm). The cell suspension from the cortex was centrifuged (500× *g*, 7 min, 4 °C) in Hanks solution (with Ca2+ and Mg2+), and the precipitate was diluted in DMEM (DMEM Low glucose (1 g/L) (PanEco, Moscow, Russia) or DMEM High glucose (4 g/L) (PanEco, Moscow, Russia)) with antibiotic-antimycotic ((Gibco, Thermo Fisher Scientific, 168 Third Avenue, Waltham, MA, USA), 0.5 mM L-glutamine (Sigma-Aldrich, St. Louis, MO, USA) and 10% heat-inactivated fetal bovine serum (Invitrogen, Thermo Fisher Scientific, 168 Third Avenue, Waltham, MA, USA). The cell suspension was dispersed into culture flasks T25 (25 cm^2^) and incubated for 12 days at 37° C, 5% CO_2_. After 5 days of cultivation in DMEM, the culture medium was replaced with a fresh medium with antibiotic-antimycotic ((Gibco, Thermo Fisher Scientific, 168 Third Avenue, Waltham, MA, USA), 0.5 mM L-glutamine (Sigma-Aldrich, St. Louis, MO, USA) and 10% heat-inactivated fetal bovine serum (Invitrogen, Thermo Fisher Scientific, 168 Third Avenue, Waltham, MA, USA)) and flasks were placed on a shaker at 200 RPM for 8 h to dissociate microglial cells. After 10 days, the monolayer of astrocytes was trypsinized and plated into 48-well plates and maintained for 4 days in DMEM. After this, the medium was changed, and the cells were used for experiments. At each change of medium, the cells were cultured only with the glucose concentration upon their preparation. This was performed to eliminate the possible reaction of cells to changes in glucose concentration.

Six hours before the experiment, the medium was replaced by the serum-free medium of the same composition and the cells remained in the serum-free medium till the end of the experiment (Figure 1). All experiments were repeated three times. Each experiment was performed on 3 independent primary astrocyte cell culture preparations. In each preparation, cells were obtained from 3 to 5 pups. In these cultures, more than 95% of the cells were positive for the astrocyte marker glial fibrillary acidic protein, and only <2% were positive for anti-Iba1 antibodies, a microglia-specific marker.

### 4.3. Immunocytochemistry

For immunostaining cells were cultivated on glass bottom dishes (MatTek, El Segundo, CA, USA). Astrocytes were fixed in 4% paraformaldehyde for 15 min. Then cells were washed once in 120 mM Na2HPO4 for 10 min, once in low-salt buffer (150 mM NaCl, 10 mM Na2HPO4) for 10 min and twice with high-salt buffer (0.5 M NaCl, 20 mM Na2HPO4) for 10 min. Cells were permeabilized and blocked with FSBB buffer (0.1% Triton X-100, 5% FBS in PBS) for 20 min and incubated overnight at +4 °C with primary anti-GFAP antibody (Sigma-Aldrich, St. Louis, MO, USA) (1:100 in FSBB), to detect astrocytes. The following day, the cells were washed once with low-salt buffer for 10 min, twice with high-salt buffer for 10 min and incubated with secondary antibody (AlexaFluor, (Thermo Fisher Scientific, 168 Third Avenue, Waltham, MA, USA) (1:500 in FSBB) for 1.5 h at room temperature. Then the cells were washed once in 120 mM Na2HPO4 for 10 min, once in low-salt buffer (150 mM NaCl, 10 mM Na2HPO4) for 10 min, twice with high-salt buffer (0.5 M NaCl, 20 mM Na2HPO4) for 10 min and incubated with the DNA-tropic dye Syto-59 (Sigma-Aldrich, St. Louis, MO, USA) (1:20,000 in 5 mM Na2HPO4) for 20 min. The cells were washed three times with 5 mM Na2HPO4 to remove the excess dye. Images were captured using a confocal laser-scanning microscope (LSM 700, Carl Zeiss, Oberkochen, Germany).

### 4.4. MTT Assay

The cell proliferation was estimated with the MTT assay. This colorimetric assay is based on the reduction of a yellow tetrazolium salt (3-(4,5-dimethylthiazol-2-yl)-2,5-diphenyltetrazolium bromide or MTT, Sigma-Aldrich, St. Louis, MO, USA) to purple formazan crystals by metabolically active cells. Astrocytes were incubated with LPS (100 ng/mL or 1 mkg/mL) in DMEM high glucose or DMEM low glucose for 48 h. After the incubation period, the MTT solution was added (the final concentration of MTT was 1 mg/mL) and incubated for 3 h (37 °C, 5% CO_2_). Then the medium was removed, and crystals were dissolved in DMSO (300 mkl/well). The plates were shaken for 10 min for the complete solubilization, and the absorbance of the samples was measured in a 96-well plate using a microplate reader iMark (Bio-Rad, Hercules, CA, USA). The wavelength to measure the absorbance of the formazan product was 570 nm and the reference wavelength was 630 nm. Six replicates for each group were used and the values were averaged. The results were transformed into the relative value to controls (control = 1).

### 4.5. Measurement of Nitric Oxide Production

NO production was measured at 48 h after incubation of astrocytes with LPS or without endotoxin (control). NO production was calculated from the amount of nitrite detected by the Griess reaction. To measure relative amounts of nitrites/nitrates in the culture fluid, the Griess reagent (a mixture sulfanilic acid with N-(1-naphthyl)ethylenediamine) (Sigma-Aldrich, St. Louis, MO, USA) was added. Its reaction with the nitrite ions leads to the formation of colored dinitrogen compounds. The Griss reagent was added into a 96-well plate (50 mkl per well), and 50 mkl of cultured astrocytes were added into each well. The plate was incubated for 15 min at room temperature. The absorbance of the nitrite-containing samples was measured in a spectrophotometric microplate reader iMark (BioRad, Hercules, CA, USA) at a wavelength of 530 nm. Six replicates for each group were used, and the values were averaged. The results were transformed into the relative value to controls (control = 1).

### 4.6. Measurement of IL-6 Production

Interleukin 6 (Il-6) is a multifunctional cytokine mediating inflammation and stress-induced responses. To verify those findings, the cell medium concentration of IL-6 was assessed 6 and 48 h after the treatment of astrocytes with LPS. Supernatants were collected, and whole-cell extracts were prepared in RIPA lysis buffer (Sigma-Aldrich, St. Louis, MO, USA) supplemented with Complete Protease Inhibitor Cocktail (Roche Diagnostics, Mannheim, Germany).

The samples of supernatants were centrifuged for 10 min (4 °C, 10,000× *g*) before the measurements to remove cell debris. Il-6 concentration was assessed with the ELISA Kit (Quantikine Rat Il-6, R&D Systems, Minneapolis, MN, USA) according to the manufacturer’s instructions. Briefly, after incubation, the cell culture medium was removed. Samples and standards were added to each well of the microplate, which was precoated with primary antibody overnight. Each well was washed and incubated with the specific enzyme-linked polyclonal antibody for 2 h. The wells were washed to remove unbound antibody-enzyme reagent and substrate solution was added to each well. After incubation for 20 min at room temperature, the enzyme reaction was stopped. The optical density of each well was measured in a spectrophotometric microplate reader iMark (BioRad, Hercules, CA, USA) at a wavelength of 450 nm. The concentrations of interleukin were determined by comparison of the optical density results with the standard curve and were normalized to the protein concentration in the corresponding cell lysate.

### 4.7. β-Hexosaminidase Assay

We hypothesized that activation of astrocytes by LPS would be accompanied by the release of β-hexosaminidase, similar to the process of LPS-induced degranulation in mast cells. After 6 or 48 h incubation of cells with LPS, the supernatant fluids were collected, and pellets were lysed in 1% Triton-X-100. β-hexosaminidase was assayed in supernatants and cell lysates. Briefly, supernatants or cell lysates were added in a ratio of 1:1 to the reaction buffer (3.5 mg of a specific chromogenic substrate—4-nitrophenyl-N-acetyl-β-D-glucosaminide (Sigma-Aldrich, St. Louis, MO, USA)/mL of 0.04 M citrate buffer) and incubated for 2 h at 40 °C and then reaction was stopped by adding the equal volume of 0.2 M Glycine-NaOH buffer (pH 10.7). The optical density of the samples was measured in a spectrophotometric microplate reader iMark (BioRad, Hercules, CA, USA) at a wavelength of 405 nm. The results were expressed as % β-hexosaminidase release.

### 4.8. Bradford Protein Assay

The total protein level in the cell lysate was estimated by the Bradford protein assay test using the Bradford reagents (Sigma). For the standard curve, a series of BSA standard solutions with dilutions from 25 to 1 mkg/mL was prepared. The triplicate reading was averaged for each standard, control or sample. 100 mkl of the sample with 100 mkl of Bradford reagent were mixed in each well of a 96-well plate and incubated for 30–40 min at room temperature. The optical density of each well was measured in a spectrophotometric microplate reader iMark (BioRad, Hercules, CA, USA) at a wavelength of 595 nm. The concentrations of protein were estimated by the calibration curve.

### 4.9. Data Analysis and Statistics

The data were expressed as mean ± SD. Results were analyzed using GraphPad InStat version 3.1 and Prism version 8 software (GraphPad, San Diego, CA, USA). The normality of data sets was assessed using the Shapiro–Wilk’s test. In order to determine the differences between the experimental conditions, the One-way Analysis of Variance (ANOVA) followed by Tukey’s multiple comparisons tests was used and the differences were considered statistically significant at *p* < 0.05. All the experiments were repeated at least three times.

## 5. Conclusions

The present study demonstrated that long-term hyperglycemia increases proinflammatory LPS-induced activation of primary rat astrocytes, and β-hexosaminidase could be a specific marker of excessive activation of astrocytes associated with exocytosis.

## Figures and Tables

**Figure 1 ijms-25-01122-f001:**
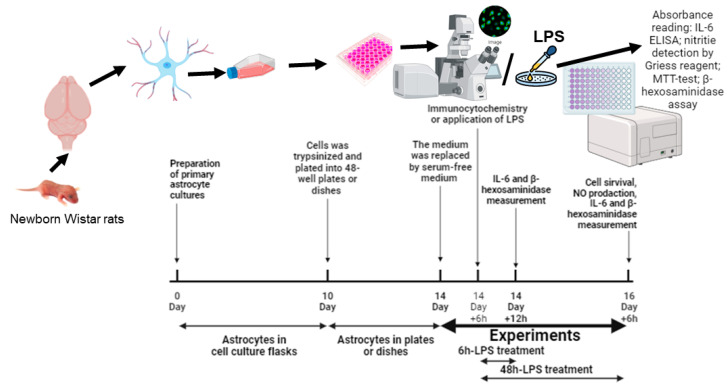
Schematic of the experiment’s timeline.

**Figure 2 ijms-25-01122-f002:**
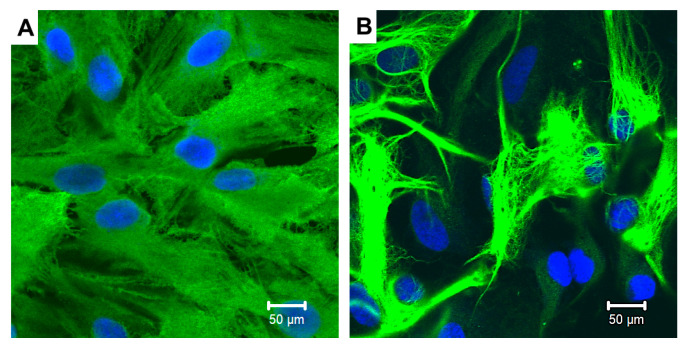
Immunocytochemistry for GFAP in cortical astrocytes cultured in a normal (**A**) or high-glucose (**B**) medium. Primary cortical astrocytes from neonate rats were cultured in normal (5.5 mM) or high-glucose (25 mM) DMEM for 14 days. Scale bar = 50 µm.

**Figure 3 ijms-25-01122-f003:**
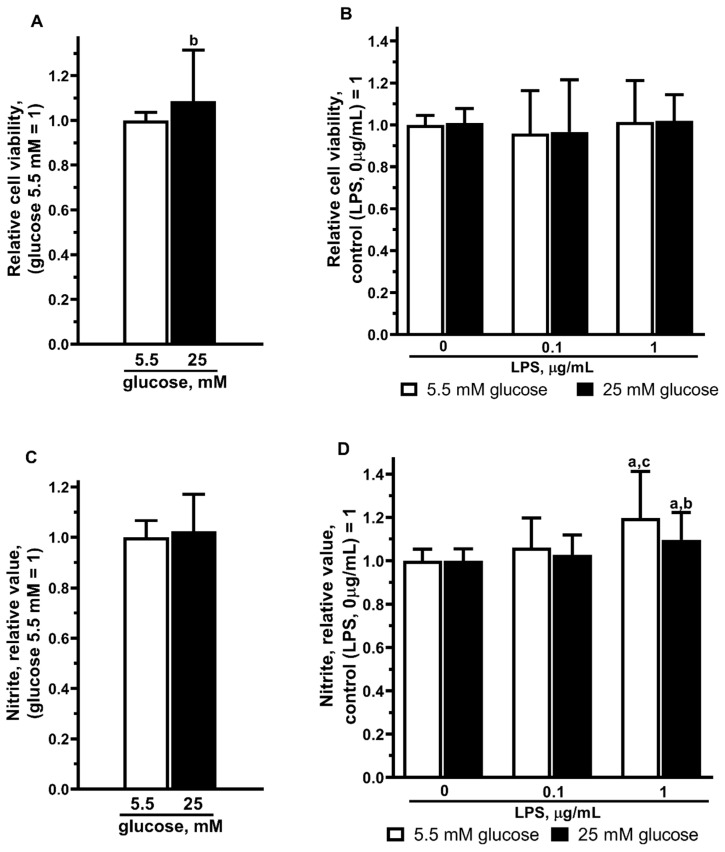
Effects of high glucose (**A**,**C**) and LPS (**B**,**D**) in medium containing NG or HG, respectively, on viability (**A**,**B**) and NO production (**C**,**D**) of primary astrocytes. Cells were cultured in DMEM with 5.5 mM glucose (NG) or 25 mM glucose (HG) for 14 days. After that, the experiment was started. The medium was replaced by serum-free medium of the same composition 6 h before the experiment, and then cells were stimulated with LPS (0.1 or 1 μg/mL) or without LPS (control). After 48 h, cell viability was detected with the MTT assay kit, the supernatant medium was collected, and nitrites were measured by the Griess reagent. Values are expressed as the mean ± S.D. of triplicate cultures. a—*p* < 0.05 vs. controls; b—*p* = 0.0031 (**A**), *p* = 0.0169 (**D**) vs. similar group in NG medium; c—*p* = 0.0001 vs. similar group with 0.1 µg/mL LPS application.

**Figure 4 ijms-25-01122-f004:**
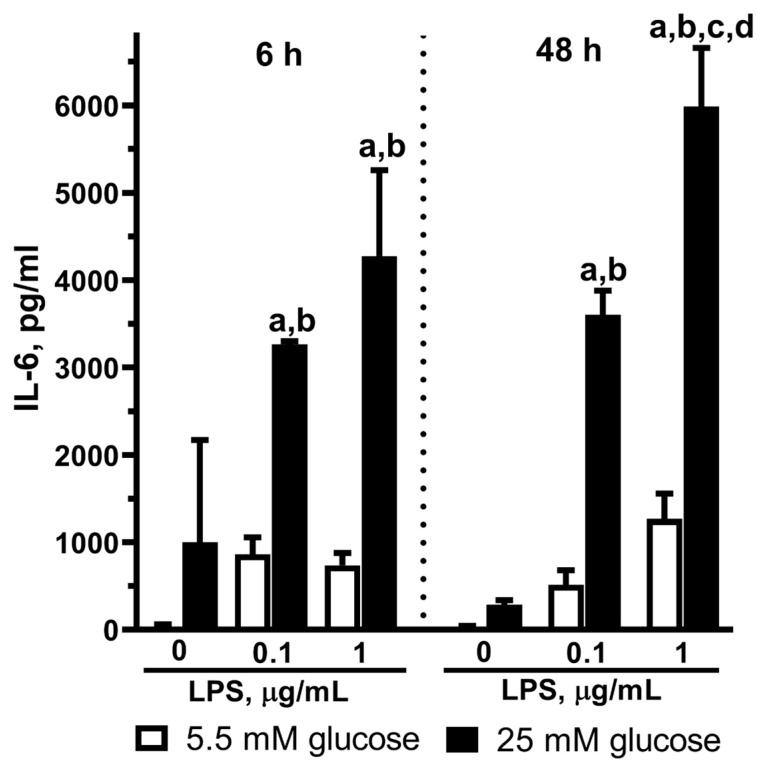
Effect of high glucose (25 mM) on the IL-6 release in primary astrocytes stimulated by LPS during 6 h or 48 h. Cells were cultured in DMEM with 5.5 mM glucose (NG) or 25 mM glucose (HG) for 14 days, after that, the experiment was started. The medium was replaced by serum-free medium of the same composition 6 h before the experiment, and then cells were stimulated with LPS (0.1 or 1 μg/mL) or without LPS (0 μg/mL LPS, control). After 6 h or 48 h, the supernatant medium was collected, and IL-6 was measured by an ELISA kit. Values are expressed as the mean ± S.D. of triplicate cultures. a—*p* < 0.05 vs. controls; b—*p* < 0.0001 vs. similar group in NG medium; c—*p* < 0.001 vs. similar group with 0.1 µg/mL LPS application; d—*p* < 0.05 vs. similar group with 6 h exposure.

**Figure 5 ijms-25-01122-f005:**
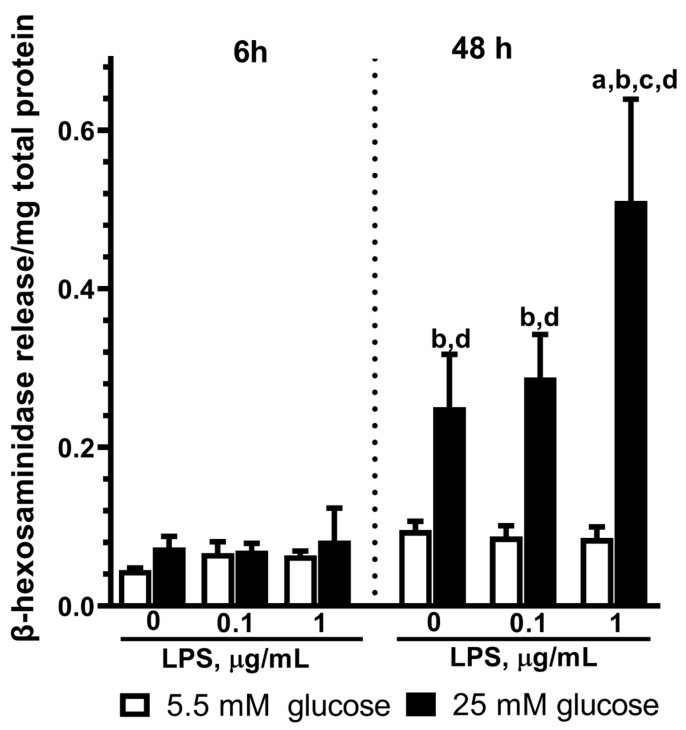
Effect of high glucose (25 mM) on the β-hexosaminidase release in primary astrocytes stimulated by LPS during 6 h and 48 h. Cells were cultured in DMEM with 5.5 mM glucose (NG) or 25 mM glucose (HG) for 14 days, after that, the experiment was started. The medium was replaced by serum-free medium of the same composition 6 h before the experiment, and then cells were stimulated with LPS (0.1 or 1 μg/mL) or without LPS (0 μg/mL LPS, control). After 6 h or 48 h, the supernatant medium was collected and β-hexosaminidase was determined using an enzymatic assay as described in Section 4.7. Values are expressed as the mean ± S.D. of triplicate cultures. a—*p* < 0.05 vs. controls; b—*p* < 0.05 vs. similar group in NG medium; c—*p* < 0.001 vs. similar group with 0.1 µg/mL LPS application; d—*p* < 0.05 vs. similar group with 6 h exposure.

## Data Availability

Data is contained within the article.

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
