# Peer review of "Long-Term Exposure of Cultured Astrocytes to High Glucose Impact on Their LPS-Induced Activation"

_ijms, 2024, doi:10.3390/ijms25021122_

Round 1

Reviewer 1 Report

Comments and Suggestions for Authors

The manuscript by Abdyeva and colleagues describes the response of primary rat astrocytes in the presence of high glucose. The authors demonstrate that, when challenged with HG, astrocytes secrete β-hexosaminidase and IL-6. Most of the results are presented clearly and easy to follow, but there are a few minor points:

  1. Is there any morphological change in astrocytes of high glucose exposure?

  2. Why did the authors not evaluate astrocyte activation by using glial fibrillary acidic protein as a marker?

  3. As a minor point, some sentences require rewriting in better English.

  4. The grant number is listed 2 times, under Funding and under Acknowledgments sections

  5. Section materials and methods - should be corrected, e.g. now Section 4.2. of methods is not filled, Section 4.3. Reagents section is duplicated again but without text, etc.

  6. There are not many papers that have studied HG on astrocytes and it makes sense to cite them, e.g. not mentioned are https://pubmed.ncbi.nlm.nih.gov/36154879/ or https://www.ncbi.nlm.nih.gov/pmc/articles/PMC7106974/ .

Comments on the Quality of English Language

As a minor point, some sentences require rewriting in better English.

Author Response

Answers on review

The authors are grateful to the reviewer for carefully reading the manuscript and making comments.

The manuscript by Abdyeva and colleagues describes the response of primary rat astrocytes in the presence of high glucose. The authors demonstrate that, when challenged with HG, astrocytes secrete β-hexosaminidase and IL-6. Most of the results are presented clearly and easy to follow, but there are a few minor points:

  1. Is there any morphological change in astrocytes of high glucose exposure?

Answer: We have performed immunicytochemistry with specific anti-GFAP antibodies in astrocyte cultures cultivated in high-glucose and normal-glucose medium and included addition Chapter 2.1. "The morphology of astrocytes cultured in normal- or high-glucose medium". The immunocytochemistry for GFAP demonstrated that astrocytes cultured in NG medium exhibited a flat polygonal shape, whereas astrocytes cultured in HG medium showed an atypical appearance,  elongated shape with long branches and with low number cell-cell contacts.

  1. Why did the authors not evaluate astrocyte activation by using glial fibrillary acidic protein as a marker?

Answer: We agree that protein GFAP expression in astrocytes can be considered their activation, but this approach is widely used in the case of the analysis of brain sections using immunohistochemistry. In this case, the proportion of immunopositive cells is assessed, which is a valid method for assessing astrogliosis.

In culture, almost all astrocytes are stained with anti-GFAP antibodies, as seen in Figure 2 in the manuscript. Thus, it is challenging to assess the expression of this protein in astrocyte culture with a high degree of reliability, and the validity of evaluating cell activation in this manner appears to be highly uncertain.

  1. As a minor point, some sentences require rewriting in better English.

Answer: We have corrected certain parts of the manuscript text.

  1. The grant number is listed 2 times, under Funding and under Acknowledgments sections

Answer: We have removed repetitions in the text.

  1. Section materials and methods - should be corrected, e.g. now Section 4.2. of methods is not filled, Section 4.3. Reagents section is duplicated again but without text, etc.

Answer: We have deleted the duplicates in Section 4.

  1. There are not many papers that have studied HG on astrocytes and it makes sense to cite them, e.g. not mentioned are https://pubmed.ncbi.nlm.nih.gov/36154879/ or https://www.ncbi.nlm.nih.gov/pmc/articles/PMC7106974/ .

Answer: We have included additional references in the manuscript.

  1. Comments on the Quality of English Language: As a minor point, some sentences require rewriting in better English.

Answer: We have rewritten some sentences, correcting the English.

Reviewer 2 Report

Comments and Suggestions for Authors

To authors: the authors use an in vitro system intended to model overexposure to glucose in astrocytes under inflammatory challenge (using LPS as stimulus) to mimic diabetes-associated inflammation. They report increased release of  inflammatory markers (IL-6 and NO) and the lysosomal enzyme β hexosaminidase (as an indicator of lysosome exocytosis), after the double challenge (LPS + 25 mM glucose) and no effects on cell survival. The results amount to a very limited set of in vitro data with distant application to diabetes. This is merely a preliminary work, too few experiments.

Specific comments.

The selection of the 3 biomarkers seems random. There are many more potential markers of astrocyte activation in response to inflammatory and glycemic challenges. Their number is so limited that it does not provide any significant information in a time where large amounts of data are readily achievable.

The discussion needs to be re-organized and re-written. Too many mixed statements and explanations. Most of the text is a series of poorly connected statements. This is in partly due to the very limited amount of new data provided.

Minor:

The sentence starting in ln 113 is out of place.

Ln 146-147: specify what you show in Fig 2, not only “the influence of glucose”

Correct typo in ln 502

Comments on the Quality of English Language

Require minor editing

Author Response

Answers on review

The authors are grateful to the reviewer for carefully reading the manuscript and making comments.

Comments and Suggestions for Authors

To authors: the authors use an in vitro system intended to model overexposure to glucose in astrocytes under inflammatory challenge (using LPS as stimulus) to mimic diabetes-associated inflammation. They report increased release of  inflammatory markers (IL-6 and NO) and the lysosomal enzyme β hexosaminidase (as an indicator of lysosome exocytosis), after the double challenge (LPS + 25 mM glucose) and no effects on cell survival. The results amount to a very limited set of in vitro data with distant application to diabetes. This is merely a preliminary work, too few experiments.

Specific comments.

  1. The selection of the 3 biomarkers seems random. There are many more potential markers of astrocyte activation in response to inflammatory and glycemic challenges. Their number is so limited that it does not provide any significant information in a time where large amounts of data are readily achievable.

Answer: The chosen research topic is the effect of hyperglycemia on astrocytes (the most numerous brain cells), which is associated with the problem of a high risk of stroke in patients with diabetes. In this regard, assessment of astrocyte activation has been considered as a possibility of potentiating astrogliosis in diabetic conditions. Astrogliosis is characterized by morphofunctional changes in astrocytes, which include parameters such as proliferation, morphological changes (data added to the new version of the manuscript), and secretory activity. In accordance with this, the studied parameters were selected: MTT test, GFAP and different types of secretory activity: nitric oxide, one of the main pro-inflammatory cytokines - IL-6 and β--hexosaminidase, as indicator of exocytosis.

  1. The discussion needs to be re-organized and re-written. Too many mixed statements and explanations. Most of the text is a series of poorly connected statements. This is in partly due to the very limited amount of new data provided.

Answer: We have rewritten and structured the discussion and corrected the results section in the manuscript.

Minor:

  1. The sentence starting in ln 113 is out of place.

Answer: We have removed the said sentence from the manuscript.

  1. Ln 146-147: specify what you show in Fig 2, not only “the influence of glucose”

Answer: We have made appropriate changes to the text of the manuscript.

  1. Correct typo in ln 502

Answer: We have fixed the error.

  1. Comments on the Quality of English Language: Require minor editing

Answer: We have rewritten some sentences, correcting the English.

Reviewer 3 Report

Comments and Suggestions for Authors

The manuscript titled "Long-term exposure of cultured astrocytes to high glucose impact on their LPS-induced activation" examines the influence of elevated glucose on responsiveness to LPS in cell culture.  The authors have designed a straight forward experiment, but there is considerable work that remains to be done before this reviewer is satisfied with the level and quality of work.  Below are some of the large issues that need to be addressed.  

1) The authors should include a schematic clearly illustrating the timeline of their experiments.  For example, it is unclear if media was harvested at day 14 or at day 16 for LPS experiments.  

2) Figures 1 and 2 should be combined as they are relatively simplistic and directly related.   Which raises another question.  Is data from Figure 1 included in Figure 2 (zero LPS treatment) or is that different data.  If different, it is unclear what figure 1 accomplishes and why a similar experiment in Figure 2 has a different outcome.  

3) Given the current layout, Results followed by Discussion, the authors should remove all discussion that appears in the Results section as that is distracting from interpretation of the data.  Alternatively, the authors could combine the Results and Discussion sections but I'll default the the journal on specifics of formatting.  

4) As the authors claim this is the first study to examine high glucose and LPS, it appears to fall short on a thorough analysis outside of cell viability, NO production, IL-6, and beta-hexosaminidase.  While the assessment of these are warranted, several other cytokines could have been examined.  

5) Given the chronic nature of these diseases, should the LPS be applied longer than just 48 hours?  This is more of a theoretical comment for authors to consider in the future or possibly provide a reason why this is not feasible.  

6) It is unclear what statistical significance is within the figures.  I suggest using letters to denote differences, which will declutter your figures and allow for immediate interpretation.  

7) In the methods section, there is template text that was not removed (Section 4.2 and 4.3).  Please thoroughly and carefully read though the manuscript to correct this error.  

8) Lines 78-98 seem out of place for that part of the introduction as by that point you should be at the level of detail you wish to examine.  This paragraph seems big picture and may need to be moved up in the introduction or into the discussion section.  

9) Astrocytes from newborn rats were utilized for this experiment.  Are there any reports on developmental changes in astrocytes?  Would they be hypersensitive to glucose compared to mature rat astrocytes?  What is the rationale for using newborn rat astrocytes? 

Comments on the Quality of English Language

Overall English is good but some reorganization of text and careful proofreading is warranted.  

Author Response

Answers on review

The authors are grateful to the reviewer for carefully reading the manuscript and making comments.

Comments and Suggestions for Authors

The manuscript titled "Long-term exposure of cultured astrocytes to high glucose impact on their LPS-induced activation" examines the influence of elevated glucose on responsiveness to LPS in cell culture.  The authors have designed a straight forward experiment, but there is considerable work that remains to be done before this reviewer is satisfied with the level and quality of work.  Below are some of the large issues that need to be addressed. 

1) The authors should include a schematic clearly illustrating the timeline of their experiments.  For example, it is unclear if media was harvested at day 14 or at day 16 for LPS experiments. 

Answer: We have included Figure 1. Schematic of the experiment's timeline in the manuscript.

2) Figures 1 and 2 should be combined as they are relatively simplistic and directly related.   Which raises another question.  Is data from Figure 1 included in Figure 2 (zero LPS treatment) or is that different data.  If different, it is unclear what figure 1 accomplishes and why a similar experiment in Figure 2 has a different outcome. 

Answer: The presentation of the graphs has been changed (please see Figure 3). To illustrate the possible effect of increased glucose in the astrocyte culture medium, data on proliferation and NO production were presented in relative values in comparison with the results obtained under conditions of normal glucose levels, which were taken as 1. At the same time, it is more appropriate to evaluate the effects of LPS in relative values in relation to control conditions (without LPS) in order to identify exactly the effect of LPS and how much this effect is determined by the level of glucose, which forces us to compare the change and not the absolute value of the measured parameters (MTT and NO).

3) Given the current layout, Results followed by Discussion, the authors should remove all discussion that appears in the Results section as that is distracting from interpretation of the data.  Alternatively, the authors could combine the Results and Discussion sections but I'll default the the journal on specifics of formatting. 

Answer: The text is structured in accordance with the content of the sections.

4) As the authors claim this is the first study to examine high glucose and LPS, it appears to fall short on a thorough analysis outside of cell viability, NO production, IL-6, and beta-hexosaminidase.  While the assessment of these are warranted, several other cytokines could have been examined. 

Answer: We agree with the reviewer that the work would have been better if a larger spectrum of cytokines had been analysed. We will fill this gap in our next study. However, in the present study, an attempt was made to comprehensively assess the state of astrocytes in the context of astroglial changes in response to the action of LPS under conditions of hyperglycemia. Therefore, parameters were selected to assess the morphofunctional changes in astrocytes characteristic of astrogliosis: MTT test (proliferation), morphological changes using anti-GFAP antibodies and different types of secretory activity: production of nitric oxide, secretion of the known marker of the pro-inflammatory response, IL-6, and exocytosis by the release of β-hexosaminidase.

5) Given the chronic nature of these diseases, should the LPS be applied longer than just 48 hours?  This is more of a theoretical comment for authors to consider in the future or possibly provide a reason why this is not feasible. 

Answer: We are grateful to the reviewer for this comment and will investigate this problem in the future.

6) It is unclear what statistical significance is within the figures.  I suggest using letters to denote differences, which will declutter your figures and allow for immediate interpretation. 

Answer: Corresponding changes to the schedules have been made.

7) In the methods section, there is template text that was not removed (Section 4.2 and 4.3).  Please thoroughly and carefully read though the manuscript to correct this error. 

Answer: We have deleted the duplicates in Section 4.

8) Lines 78-98 seem out of place for that part of the introduction as by that point you should be at the level of detail you wish to examine.  This paragraph seems big picture and may need to be moved up in the introduction or into the discussion section. 

Answer: The error has been corrected, and the text has been removed.

9) Astrocytes from newborn rats were utilized for this experiment.  Are there any reports on developmental changes in astrocytes?  Would they be hypersensitive to glucose compared to mature rat astrocytes?  What is the rationale for using newborn rat astrocytes?

Answer: The primary culture of astrocytes used in this work is a generally accepted and well-validated method that allows us to evaluate the reactions of astrocytes under different conditions. It has been shown that the isolation of astrocytes from the brain of newborn animals can significantly increase the safety and survival of cells during the procedure of mechanical suspension and proteolytic digestion of brain tissue, which is not observed when isolating astrocytes from the brain of adult animals, where they are already highly specialized and differentiated.

Comments on the Quality of English Language

Overall English is good but some reorganization of text and careful proofreading is warranted. 

Answer: Certain parts of the text have been reorganized, and the text has been proofread.

Round 2

Reviewer 3 Report

Comments and Suggestions for Authors

I thank the authors for the edits and explanations.  I now recommend for publication.